# Effects of Surfactants on Zein Cast Films for Simultaneous Delivery of Two Hydrophilic Active Components

**DOI:** 10.3390/ma15082795

**Published:** 2022-04-11

**Authors:** Dongwei Wei, Fanhui Zhou, Hongdi Wang, Guijin Liu, Jun Fang, Yanbin Jiang

**Affiliations:** 1School of Chemical Engineering and Materials Science, Quanzhou Normal University, Quanzhou 362000, China; wdw2017@qztc.edu.cn (D.W.); jingxu@qztc.edu.cn (F.Z.); 2Key Laboratory of Organosilicon Chemistry and Material Technology, College of Material, Chemistry and Chemical Engineering, Ministry of Education, Hangzhou Normal University, Hangzhou 311121, China; wang.hongdi@hznu.edu.cn; 3School of Pharmaceutical Sciences, Hainan University, Haikou 570228, China; liugj@hainanu.edu.cn; 4School of Chemistry and Chemical Engineering, South China University of Technology, Guangzhou 510640, China

**Keywords:** zein, surfactant, lysozyme, ascorbic acid, controlled release, emulsified mechanism

## Abstract

In order to prepare edible films with outstanding antimicrobials and antioxidants utilized in applications of food and pharmaceutics, in this study, effects of surfactants on zein cast films for simultaneous delivery of lysozyme (LY) and ascorbic acid (AA) were investigated, where sodium alginate (SA), soy lecithin (SL), and Pluronic f-68 (PF-68) were selected as surfactants. FT-IR tests indicated that SL or PF-68 dramatically changed secondary structure of zein composite films, which heightened the irregularity of the composite film and inhibited LY crystallization. Mechanical tests showed that highly flexible films exhibiting elongations between 129% and 157% were obtained when adding PF-68. Compared with the film without emulsifier, zein film containing SL and PF-68 showed approximately 7.51 and 0.55 times lower initial release rates for LY and AA respectively, which significantly improved the controlled release and heightened the anti-microbial and anti-oxidant activities of the film. Finally, emulsified mechanisms of the surfactants in zein films were proposed.

## 1. Introduction

Recently, investigation of edible films containing antimicrobials and antioxidants utilized in applications of food and pharmaceutics has gained massive interest due to the promising results, not only prolonging the shelf time but also improving the quality of the products, controlling the release and alleviating the toxicity of pharmaceutics through their applications [1,2]. Among of them, zein, the major co-product of the oil and rapidly growing bioethanol industries, has drawn dramatic attention in the applications of food and pharmaceutics [3]. Zein is a water-insoluble prolamin present in corn endosperm cells with a high content of hydrophobic amino acids (leucine, proline, and alanine) [4]. Due to the good filming property [5], nontoxic [6], excellent biocompatibility [7], extensive sources [8] and low costs [9], dramatic investigations of zein utilized as carries in food and pharmaceutics have been conducted [10].

However, pure zein film has its intrinsic flaws, e.g., poor mechanical properties and not good enough controlled release properties. To overcome the shortcomings, dramatic investigations for modifying zein films with various materials were conducted, e.g., small organic molecules [11], synthetic polymer [12], saccharides [13], lipids [14], and proteins [15]. Among of the various materials, surfactant is a kind of feasible option. In biology, chemistry, and pharmaceutics, surfactants are commonly used in upstream and downstream processing of therapeutic proteins, which modulate adsorption loss and aggregation by coating interfaces and/or participating in protein–surfactant association. Therefore, many investigations about the mechanisms and interaction of proteins and surfactants in solutions are conducted [16,17]. But the effects of surfactants on proteins in solutions are eventually reflected in protein products with solid form, e.g., nanoparticles, films, and capsules. Thus, the investigation for the effects of surfactants on protein products is also very important. However, only fewer studies were conducted, e.g., Chuacharoen et al. investigated the effects of lecithin and pluronic F127 surfactants on the stability and controlled release of lutein loaded in zein nanoparticles [18]. Li et al. investigated the effects of Pluronic F127 on the structure and physical properties of zein composite films [19], and Xiao et al. studied the release kinetics of nisin in spray-dried zein capsules [20]. Among the surfactants, sodium alginate (SA, anionic surfactant), soy lecithin (SL, an ampholytic surfactant), and Pluronic f-68 (PF-68, a non-ionic surfactant) are attractive. SA is a polysaccharide occurring in large amounts in nature [21], which has widespread applications in the food, drinks, pharmaceutical, and bioengineering industries [22,23]. SL, with an average molecular weight of 325.3 Da, is widely used in the food industry as an emulsifier, which has “generally recognized as safe” (GRAS) status in foods with no limitation other than current good manufacturing process [24]. PF-68 is a synthetic non-ionic surfactant with an average molecular weight of 8.4 kDa and widely used in the industrial, cosmetics, drugs, and biological domains [25,26].

The aim of the current work is to develop zein/surfactant cast films for simultaneous delivery of two hydrophilic active components, and investigate the effects of different surfactants on the simultaneous release profiles of the model active components and emulsified mechanisms of surfactants in the films, where SA, SL, and PF-68 were selected as model surfactants, antimicrobial agent—lysozyme (LY) [27,28] and antioxidant agent—ascorbic acid (AA) [29,30] were selected as hydrophilic active components. This study will probably provide a scientific basis for investigating the delivery system for simultaneous release of hydrophilic active components, which has the potential applications in food, bioengineering, and pharmaceutics.

## 2. Materials and Methods

### 2.1. Materials

AA and Trolox (99.99 wt%) were purchased from Aladdin Ind. Corp (Shanghai, China). Zein (Food Grade Quality) was purchased from Sigma-Aldrich Shanghai Trading Co. Ltd. (Shangai, China) LY, SA, SL, PF-68, *Listeria innocua* (*L. innocua,* S25897) and *Micrococcus lysodeikticus* (Biological reagent) were purchased from the Shanghai Yuanye Bio-Technology Co. Ltd. (Shangai, China) PEG 400, ethanol (Et) (Analytic Reagent) were purchased from the Guangdong Guanghua Sci. Tech. Co., Ltd. (Shantou, China). All other chemicals were reagent grade.

### 2.2. Preparation Procedure of Zein Film

The cast films were produced by modifying the method described in Soliman et al. [31], described as follows. (1) 1.2 g of PEG 400, 30 mL of 80% v/v aqueous ethanol, and 3 g of zein were mixed and the film-forming solutions were obtained. (2) The film-forming solutions were stirred for 20 min and treated by an ultrasonic instrument for 40 s (s). (3) The solutions were heated at 70 °C for 10 min at a stirring speed of 500 rpm and cooled to room temperature. (4) The surfactants were added by stirring and their concentrations were expressed as percentages with respect to the mass of zein. After dissolution of the surfactants, LY (16.5 mg/g film-forming solution, 20,000 U/mg protein) and AA (12.5 mg/g film-forming solution) were then added into film-forming solutions (the final LY and AA concentrations in dried films were 4.1 and 3.1 mg/cm^2^). (5) 10 mL of the film-forming solution was poured into rectangular polypropylene vessels and dried at 30 °C and 30% relative humidity (RH) for 36 h (h). (6) After drying, a micrometer was used to determine the average thickness of the films by measuring the thickness at different points of each film (at least 6 times). (7) 2 × 8 cm^2^ wide strips were obtained by cutting the sample films and utilized in the tests.

### 2.3. Characterization Methods

#### 2.3.1. Scanning Electron Microscope

The microstructure of the selected cast films was observed and obtained by a scanning electron microscope (SEM) including surface and cross-section morphology (Zeiss MERLIN Field Emission SEM, Carl Zeiss NTS GmbH, Oberkochen, Germany).

#### 2.3.2. Fourier Transform Infrared Spectra

Fourier transform infrared spectra (FT-IR) of the selected samples were obtained using a FT-IR spectrophotometer (Nicolet Nexus 670, Thermo Electron Corporation, Waltham, MA, USA) by adding the film-forming solutions onto the surface of a tablet containing approximately 20 mg of KBr and drying for 10 min. The measurements were carried out in the wave number region of 400–4000 cm^−1^. To investigate the effects of surfactants on the secondary structure of zein composite films, deconvolution of each spectrum was performed using Peak Fit software according to the methods of Fourier self-deconvolution (FSD) and the assignment of individual components to the secondary structural elements refers to Mizutani et al. [32].

#### 2.3.3. Mechanical Properties

Mechanical properties of the developed zein cast films, i.e., tensile strength at break (TS), elongation at break (EB), and Young’s modulus were determined using a universal testing machine (3367, INSTRON, Norwood, MA, USA) according to the ASTM standard method D882-02 [33]. Before tests, all the films were kept for equilibrium at the conditions of 50% RH and 35 °C at least 48 h. The initial grip distance and crosshead speed of the testing machine was 50 mm and 50 mm/min respectively. At least 6 replicates of each kind of films were tested.

#### 2.3.4. Differential Scanning Calorimetry

The thermal properties of the typical zein-based films were determined by a differential scanning calorimetry (DSC) (Q200, TA Instruments, New Castle, DE, USA). The scanning rate and compressed nitrogen purge were set as 10 °C/min and 25 mL/min, individually. The heated temperature sets from −50 to 200 °C for the first and second circle at the same scanning rate.

### 2.4. LY Release Profiles and Anti-Microbial Potential of Films

The LY release profiles of films were proceeded by following the method described in Arcan et al. [34]. Briefly, the release tests were conducted in a constant temperature of 4 °C and at 800 rpm. The films (2 × 8 cm^2^) were placed into 150 mL of beakers containing 50 mL of distilled water. The LY activities were determined at 660 nm by using a Shimadzu spectrophotometer (Model 2450, Tokyo, Japan) at a constant temperature of 30 °C. To determine the anti-microbial potentials of the cast films, it was tested by following the method of Arcan et al. [34] and Moradi et al. [35] with minor modifications. The diameter of test films was 11 mm. The concentration of bacterial suspensions was adjusted to approximately 1.0 of optical density (OD) using a UV-vis spectrophotometer at 600 nm. About 200 μL of the bacterial inoculum was added and spread evenly onto a Petri dish. Each three round films were placed into a dish, the dishes were placed at 37 °C for 12 h, and the clear inhibition areas were examined. The final anti-microbial activities of the films were obtained to calculate inhibition areas (mm^2^) by subtracting the inhibitory area from the area of the applied film discs.

### 2.5. LY Activities in Different Surfactant-Containing Solutions

As LY activities were probably affected by the surfactants and these effects further affected the release and anti-microbial profiles of zein cast films, LY activities in solutions containing different surfactants (Categories 1–7) were determined. Briefly, solutions containing surfactants were prepared by adding 1.2 g of PEG 400, 30 mL of Et/H_2_O (80% v/v in water), 31 mg of LY (20,000 U/mg protein) (PLY), and different surfactants (SA, SL, PF-68, SA/SL, SA/PF-68 and SL/PF-68). The categories of the solutions included (1) PLY-SA (0.1 g), (2) PLY-SL (0.1 g), (3) PLY-PF-68 (0.1 g), (4) PLY-SA (0.05 g)-SL (0.05 g), (5) PLY-SA (0.05 g)-PF-68 (0.05 g), (6) PLY-SL (0.05 g)-PF-68 (0.05 g), (7) PLY and (8) PEG 400 (1.2 g)-LY (31 mg)-H_2_O (30 mL).

### 2.6. AA Release Profiles and Anti-Oxidant Capacity of Films

AA release properties of the obtained films were conducted by following the method described as the process of LY release tests with minor modifications. The soluble AA content of the collected samples was determined spectrophotometerically at 264 ± 1 nm according to the colorimetric method of Cupello et al. [36,37]. All concentration measurements were conducted three times.

### 2.7. Statistical Analysis

To determine the effect of surfactants on mechanical properties, anti-microbial, and anti-oxidant potentials of developed films, analysis of variance (ANOVA) was applied, the mean values were subjected to Duncan’s test, and a *p*-value of <0.05 was considered statistically significant using the software IBM SPSS Statistic 19. All tests were repeated at least three times and the results were reported as averages and standard deviations of these measurements.

## 3. Results and Discussion

### 3.1. Microstructures

The SEM images of surfaces (Appendix A) and cross-sections (Figure 1) were conducted to investigate the microstructures of developed zein cast films. The film without surfactant (Film 2) clearly indicated that there were many LY aggregates existing within the film. For the films containing single emulsifier, the film containing SA (Film 4) showed that the surface was smooth, and the cross-section was incompact and porous. The film containing SL (Film 7) had impact and smooth microstructure, but contained many larger LY aggregates, which was attributed to the increased hydrophobic property because of the addition of SL. For the film containing PF-68 (Film 10), a lot of nanoparticles (PF-68 crystals) with diameters of approximately 100 nm existed on the surface [19]. The PF-68 crystals will be further proved in the DSC tests. PF-68 is a hydrophilic non-ionic surfactant co-polymer consisting of a hydrophobic block of polypropylene (PPO), which was located between two hydrophilic blocks of polyethylene glycol (PEO) [38]. Hydrophilic PF-68 will crystalize in hydrophobic zein matrix and the crystals locating on the surface can be attributed to three factors, i.e., density, surface tension, and hydrophobic interactions. Because density of zein is estimated to be >1.36 g/mL [39] and the density of PF-68 is about 1.05 g/mL, PF-68 crystals are prone to floating on the surface of zein-based film. Besides, because of the hydrophilic and self-crystallization of the PEO block, it has lower miscibility than PPO block to zein, and tends to segregate from the hydrophobic regions of zein and PPO block, and enriches the film surface. Meanwhile, the surface tension of PEO is larger than that of PPO and zein, which also promote the segregation of PEO blocks onto the air/solid interface. Li et al. [19] and Johnson et al. [40] both found the similar phenomena in their studies.

For the films containing multiple surfactants, it was clearly shown that the film containing SA and SL (Film 12) had a smooth surface and compact cross-section, and the film containing SA and PF-68 (Film 13) had clear cracks on the surface which could lead to rapid release of LY and AA. For the film containing SL and PF-68 (Film 14), it could be seen that the surface is smooth, and small PF-68 nanocrystals were found in the cross-section, which were well dispersed within zein matrix. The results indicated that compared with the film containing PF-68, addition of SL heightened the dispersion of PF-68 nanocrystals in zein matrix.

### 3.2. FT-IR Characterization

The FT-IR spectra of zein, LY, AA, PEG 400, PF-68, SA, and SL are shown in Appendix A, and the developed films (Films 2, 4, 7, 10, 12, 13 and 14) are shown in Appendix A. As shown in Appendix A, the absorbance peaks of 1647.1 cm^−1^ and 1649.0 cm^−1^ (amide I) identified as C=O stretching were selected as characteristic absorptions in studying zein and LY respectively. As shown in Appendix A, the results indicated that taken as a whole, no new peak appeared which showed that no chemical reaction occurred. However, it was clearly shown that shifts in characteristic absorptions for zein and LY all occurred in the films. The characteristic absorptions of zein and LY in the films for Films 2, 4, 7, 10, 12, 13, and 14 shifted approximately from 1647.1 cm^−1^ to 1654.8 cm^−1^, 1656.8 cm^−1^, 1660.6 cm^−1^, 1660.6 cm^−1^, 1658.7 cm^−1^, 1662.6 cm^−1^, and 1660.6 cm^−1^, and the blueshifts are shown in Figure 2. The results showed that the synergistic effects of PEG 400 and AA played a dominant role in promoting the blueshifts occurrence. It could also be seen that all the surfactants heightened the blueshifts, which suggested that the surfactants decreased the hydrogen bonds of zein and LY, and heightened the dissolvability of LY in the films.

To further investigate the effects of surfactants on the secondary structure of zein composite films, the Fourier deconvolution of Films 2, 7, 10, and 14 were conducted and shown in Figure 3 and Table 1. The results clearly showed that compared with Film 2, the content of α-hellix (regular conformation) dramatically decreased and the content of β-turn (irregular conformation) significantly increased after the addition of surfactants (Films 7, 10, 14). It indicates that surfactants can significantly change the secondary structure of proteins (zein and LY), and suggests that conformation of protein molecules transfers from regular to irregular situation, which induces the aggregation of proteins [41]. It could be attributed to that the hydrophobic parts of SL and PF-68 affect the hydrophobic side chains of α-hellix, and the hydrophilic parts affect the formation of hydrogen bonds of α-hellix, which decrease the content of α-hellix in the composite films. Similar phenomena were also found by Qi et al. [41] and Duodu et al. [42], who concluded that protein tended to form more intermolecular β-sheet structure perhaps at the expense of some α-helix conformation, and the change in the secondary structure of the protein was due to heat denaturalization and aggregation. Besides, it also can be seen that SL and PF-68 basically have no different effects on the secondary structure of proteins.

### 3.3. Mechanical Properties of Films

The mechanical properties of zein cast films including the parameters of TS, EB, Young’s modulus, and film thickness are measured and listed in Table 2. The results of the film containing LY and AA (Film 2) indicated that compared with the film without LY and AA (Film 1), the addition of AA and LY significantly decreased TS by 38.2% (*p* < 0.05) and accordingly increased the EB by 190%. The results were mainly due to the synergistic effect of AA and PEG 400 that had been discussed in our previous study in detail.

Compared with the film without surfactants (Film 2), the mechanical test results of the films containing SA (Films 3, 4 and 5) showed that the addition for 5% SA did not change the TS, but dramatically increased the EB (*p* < 0.05) and accordingly decreased Young’s modulus (*p* < 0.05). In contrast, the addition for 10% of SA significantly increased the TS (*p* < 0.05), but did not change the EB value (*p* > 0.05) and dramatically increased the Young’s modulus accordingly (*p* < 0.05). When the concentration of SA reached 15%, the TS value was dramatically increased, but the EB value was apparently increased too and accordingly Young’s modulus was not changed significantly. It could be concluded that the concentration of SA in zein cast films dramatically affected the mechanical properties of the films. The effect could be attributed to the hydrophilic property and long-chain structure of SA. For the films containing SL (Films 6, 7, and 8), it was shown that compared with the film without surfactants (Film 2), the addition of SL (5, 10 and 15%) significantly increased the TS values (*p* < 0.05), decreased the EB values (*p* < 0.05), and accordingly increased Young’s modulus (*p* < 0.05). The results indicated that the addition of SL weakened the flexibility of the zein cast film. SL is hydrophobic and a phospholipid food emulsifier or stabilizer with a hydrophilic head, phosphatidylcholine (PC) and two hydrophobic tails, phosphatidylethanolamine (PE) and phosphotidylinositol (PI) [43]. Because the hydrophilic head PC is higher than that of PEG 400, the addition of SL breaks the synergistic effects of PEG 400 and AA on strengthening the plastification of zein cast films.

For the films containing PF-68 (Films 9, 10 and 11), the results indicated that the addition of PF-68 did not significantly change the TS (*p* > 0.05), but dramatically increased the EB values (*p* < 0.05) and accordingly significantly decreased Young’s modulus (*p* < 0.05). The dramatical increase of the EB values could be due to the hydrophilic segments of PF-68. It is clearly shown that PF-68 can be used as a good plasticizer incorporating with PEG 400. Meanwhile, the result also demonstrates that although many nanocrystals of PF-68 are formed onto the surface of Film 10, parts of the added PF-68 still exist in the zein composite film and effectively plasticize the film, which was attributed to the partial miscibility of zein and PF-68 at the amorphous region [19]. For the films containing multiple emulsifiers (Films 12, 13 and 14), TS, EB, and Young’s modulus values indicated that the addition of the multiple emulsifiers did not dramatically change the mechanical properties of zein cast film.

### 3.4. DSC Characterization

To determine the effects of surfactants on the thermal behaviors of zein/surfactant films, the DSC thermograms of Films 2, 7, 10, and 14 are obtained and demonstrated in Figure 4. In Film 2, a blunt peak at about 48 °C, i.e., the melting point (*T*_m_) in zein composite film, showed that LY crystals were formed during the preparation of the film. Therefore, LY aggregations in the inner parts of Film 2 existed in the form of crystals not amorphism. But the blunt peak also suggested that the crystallization of LY is not complete due to the high molar weight. The DSC curve of Film 7 indicated that LY crystals disappeared, which suggested that the addition of SL transformed the LY crystals into amorphism. The result was also demonstrated by the change of secondary structures of zein composite films. In Films 10 and 14, it is shown that the films contained PF-68 crystals not LY crystals, which can be interpreted as that the melting point of PF-68 is approximately 51 °C [44], and PF-68 has the same effect on the change of secondary structure for LY as SL does. Besides, the peak of DSC curve in Film 2 is blunt, but the peaks of Films 10 and 14 are very sharp, which suggest a new kind of crystal.

Compared with the glass transition temperature (*T*_g_) of Film 2, it was shown that the addition of SL does not significantly change the *T*_g_ of Film 7, but *T*_g_ of Film 10 increases by approximately 5 °C with the addition of PF-68. The increase of *T*_g_ also suggested the partial miscibility of zein and PF-68. According to the theory of macromolecule, existence of crystalline parts in a semi-crystalline polymer will restrict the motion of amorphous parts. Therefore, Film 10 can be considered as a blending polymer and the crystalline PEO blocks will restrict the motion of other amorphous parts, and *T*_g_ of Film 10 increases with the addition of PF-68.

### 3.5. LY Activities in Different Surfactants Contained Aqueous Ethanol

LY activities in Et/H_2_O solutions containing different surfactants are conducted and demonstrated in Figure 5. It was clearly shown that LY activity in distilled water (Category 8) was about 2–2.5 times than that in aqueous ethanol (Categories 2–7) (*p* < 0.05), which indicated that the addition of ethanol effectively inhibited LY activity. From the results of LY activities in the solutions of Categories 2–7, a conclusion was obtained that the addition of surfactants in aqueous ethanol did not significantly affect the LY activities (*p* > 0.05). However, compared with the others, SA (Category 1) significantly decreased the LY activity (*p* < 0.05), and LY activity could not be recovered during the preparation of zein cast films, which further weakened the anti-microbial activity of the films containing SA. As an anionic surfactant, the mechanism of SA denaturing LY was similar to SDS denaturing proteins [45,46]. SA is amphiphilic containing both a polar head group and a hydrophobic tail on the same molecule [47]. The charged head group can either bind electrostatically to the oppositely charged amino group of the protein, or the alkyl chain can interact through hydrophobic bonding to the non-polar groups either on the surface or the interior of the globular protein. In our study, LY is positively charged due to its high pI at 11.35 and easily bonded to SA on the surface or the interior.

### 3.6. Effect of Single Surfactant on LY Release Profiles

In order to determine the effects of single surfactant on LY release profiles, the release profiles of LY from zein cast films and kinetic parameters determined from release curves of LY are conducted and demonstrated in Appendix A. It was clearly shown that the film without emulsifier (Film 2) had the highest initial release rate and total released LY, which demonstrated that the addition of surfactants significantly sustained the release of LY from zein cast films (*p* < 0.05). Moreover, the total released LY was 41574 U/cm^2^, and the concentration of LY added in the film was 82,000 U/cm^2^, which demonstrated that approximately 50.7% of LY existed in soluble form in the film. For the films containing SA (Films 3–5), the release profiles indicated that exactly as interpreted in the former part, SA could effectively denature LY, and LY activity could not be recovered after ethanol evaporated. Therefore, zein films containing SA had the lowest initial LY release rate and total released LY (*p* < 0.05). For the films containing SL (Films 6–8) or PF-68 (Films 9–11), compared with the film without surfactant (Film 2), the initial LY release rate and total released LY both significantly decreased by about 50% (*p* < 0.05). The results clearly indicated that SL and PF-68 could both emulsify the film-forming solutions and significantly sustain the release of LY. Meanwhile, it could be seen that there was no distinction between the sustained effects of SL and PF-68 on the release of LY.

### 3.7. Effects of Multiple Emulsifiers on LY Release Profiles

To determine the effects of multiple emulsifiers on LY release profiles, the released profiles of LY from zein cast films containing multiple emulsifiers and the kinetic parameters determined from release curves of LY were conducted and shown in Figure 6 and Appendix A. For the films containing SA and SL (Film 12), the initial LY release rate and total released LY did not significantly change (*p* > 0.05) compared with the films containing SA (5% and 10%), which indicated that the addition of SL did not alleviate the denaturation of LY caused by SA. The film containing SA and PF-68 had a lot of cracks on the surface (Film 13 of Appendix A). Therefore, the film was smashed during the tests of LY release, and the total LY released was not obtained, as shown in Appendix A and Figure 6. For the films containing SL and PF-68 (Film 14), compared with Film 2, the initial LY release rate decreased by approximately 88.3% and the total released LY also decreased by 57.7% (*p* < 0.05). The results demonstrated that compared with the films containing single emulsifier (Films 6–11), the multiple emulsifiers of SL and F-68 had the most significant effect on controlling the release of LY (*p* < 0.05).

### 3.8. Effects of Surfactants on AA Released Profiles

The effects of single emulsifier and multiple emulsifiers on the AA release profiles are shown in Figure 7, and the kinetic parameters determined from release curves of AA in Figure 7 are listed in Appendix A. It was very clear that the initial AA release rate and total released AA from zein cast films significantly reduced by 2.7–43.9% and 8.0–45.6% respectively after surfactants were added (*p* < 0.05). Compared with the film without emulsifier (Film 2), the films containing SL (Films 6–8) had the lowest initial release rate and total released AA, which reduced by 43.9% and 45.6% individually and suggested that the SL is more effective to be utilized to control the release of AA. The initial release rate and total released AA of the films containing PF-68 (Films 9–11) were both higher than those of the films containing SL (Films 6–8), which indicated that SL was more effective than PF-68 to emulsify AA. The result could be attributed to the more powerful hydrophilic head of SL. It could be seen that compared with the initial AA release rate of the film containing SL, the addition of PF-68 in the film containing SL and PF-68 (Film 14) did not significantly change the initial AA release rate (*p* > 0.05), but significantly increased the total released AA by 12.9% (*p* < 0.05). The results clearly showed that the addition of SL and PF-68 could cause a great degree of reduction in the initial AA release rate (37%) and a minor reduction in the total AA released (16%). Therefore, the multiple surfactants of SL and PF-68 could be added to the zein cast film to control the release profiles of AA, and the released profiles probably could be modulated by adjusting the ratio of SL and PF-68.

### 3.9. Anti-Microbial and Anti-Oxidant Potentials of Films

The results of anti-microbial tests of the cast films on *L. innocua* are shown in Figure 8. It was shown that all the films had a clear anti-microbial zone area around the films, which indicated that the developed zein cast films had good anti-microbial effects on *L. innocua.* Although the film containing 10% PF-68 (Film 10) had a lower initial release rate and total released LY than the film without emulsifiers (Film 2) (*p* < 0.05), there were no significant differences in the zone area (*p* > 0.05), in which both had the highest anti-microbial activities. It can be seen that the film containing SL and PF-68 (Film 14) had a significantly lower initial release rate than the film containing SL or PF-68 (Films 6–11) (*p* < 0.05), but this did not cause any significant change in anti-microbial activities (*p* > 0.05). Therefore, the film emulsified by SL and PF-68 both had good controlled release properties and the same anti-microbial activity with the film containing SL or PF-68. The films containing SA (Films 3–5, 12) had the lowest anti-microbial activities because of the LY denaturation (*p* > 0.05). Therefore, it can be concluded that to prepare zein films with good controlled release properties and anti-microbial activity, it is not appropriate to add ionic surfactant (i.e., SA) in zein films, but the ampholytic surfactant (SL), non-ionic surfactant (PF-68) and a mixture of them is an alternative option.

The anti-oxidant activities of films are shown in Appendix A. It was evident that in general, the anti-oxidant activity was positively correlated with the total released AA except for the films containing SL (Films 6–8). Although the values of total released AA of the films containing SL were significantly lower than the film without emulsifier (Film 2) (*p* < 0.05), but the anti-oxidant activity was much higher than that of Film 2 (*p* < 0.05), which was due to the anti-oxidant activity of SL. Compared with the other films, although the film containing SL and PF-68 (Film 14) had the minimum initial release rate, it had the maximum value of the anti-oxidant activity (*p* < 0.05), which was up to 226.0 μmol Trolox/cm^2^. It was clearly shown that compared with Films 7 and 10, although Film 14 had the same content of surfactants, the synergistic effects of SL and PF-68 significantly controlled the release of AA and strengthened the anti-oxidant activity of the film (*p* < 0.05).

### 3.10. Emulsified Mechanisms of Surfactants in Zein Matrix

Basing on the above characterizations of zein cast films, the emulsified mechanisms of SL and PF-68 on LY and AA in the films are proposed and demonstrated in Figure 9. The emulsified mechanism of SL in the zein film is shown in Figure 9B, which indicates that because SL and zein are both hydrophobic, many relatively large LY aggregates exist, and the emulsified LY aggregates by SL are stable and distributed in the film (Film 7 of Figure 1). Hydrophobic chains of SL interact with zein chains and the hydrophilic head interacts with hydrophilic LY and AA, which forms the micellar structure and sustains release of LY and AA. Meanwhile, as an ampholytic surfactant, the hydrophilic head has both negative and positive ions, which has stronger ability to attract AA than PEG 400, and results in the weakness of the synergistic plasticization of PEG 400 and AA on the zein cast film to decrease the EB of the film (Films 6–8 of Table 2). Furthermore, the zein cast film emulsified by SL has a low release rate and total release of LY and AA (Films 6–8 of Appendix A).

The emulsified mechanism of PF-68 on LY and AA in zein/PF-68 film is demonstrated in Figure 9C. From the result of DSC tests and the references of Li et al. [19] and zhang et al. [48], it can be proposed that the structure of PF-68 in zein film is similar to the crystal structure of PF-127 in zein film. Just as the demonstration of Figure 9A, hydrophilic PEO chains of PF-68 in a hydrophobic zein domain-dominant space will be crystallized in regular arrangement. Because of the high molar weight of PF-68, the PPO block works with zein domains not zein chains [9]. As shown in Figure 9C, the hydrophobic PPO block interacts with zein domains, and the distributed PEO crystallites cannot compete with the plasticizing effect provided by PPO-zein interaction. Excessive PF-68 will be crystallized and floated onto the surface of zein/PF-68 composite film (Film 10 of Figure 1). Moreover, the addition of PF-68 sustains the release rate and total release of LY and AA (Films 9–11 of Appendix A). However, compared with the release rate and total release of LY and AA from the zein cast film containing SL (Films 6–8 of Appendix A), the release rate and total release of LY and AA from the film containing PF-68 (Films 9–11 of Appendix A) significantly increased (*p* < 0.05), especially for AA. The result is attributed to the stronger hydrophilic head of SL to interact with LY and AA, and the dissolved profile of PF-68 in water, which promotes the release rate and increases the total release of LY and AA.

The emulsified mechanism of SL and PF-68 on LY and AA in zein-SL-PF-68 composite film is shown in Figure 9D. Compared with PF-68, SL with lower molar weight has the priority to interact with zein chain with hydrophobic head, and meanwhile interact with AA and LY with hydrophilic head. The hydrophobic PPO block interacts with zein domain containing micelles (Figure 9B), which further sustains the release rate of LY and AA. Thus, it can be seen that the difference of molar weights in SL and PF-68 is a very important factor to sustain the active components. In contrast with the results of mechanical tests for the films containing SL (Films 6–8 of Table 2) or PF-68 (Films 9–11 of Table 2), the EB of the film containing SL and PF-68 (Film 14) locates between that of the two films. The result indicates that although SL is dominant in emulsifying AA, which weakens the EB of the film, the interaction of hydrophobic PPO block with zein domain dramatically increases the EB value. Therefore, compared with TS and EB of the film without emulsifier (Film 2 of Table 2), the TS and EB of the film containing SL and PF-68 (Film 14 of Table 2) do not significantly change (*p* > 0.05). Therefore, it can be seen that zein cast films with good controlled release properties for LY and AA can be developed by adding SL and PF-68.

## 4. Conclusions

Zein/surfactant cast films for simultaneous delivery of LY and AA were successfully developed and the effects of surfactants on the release profiles of the films were investigated in detail. The mechanical tests indicated that SL dramatically weakened the flexibility of the cast film, but PF-68 was just the opposite. The release results and activity tests showed that SA denatured the activity of LY, but SL and PF-68 both significantly sustained the release of LY and AA. Meanwhile, it was clear that the simultaneous addition of SL and PF-68 in the cast film significantly sustained the release of LY and AA, heightened the anti-oxidant and anti-microbial activities, and the change of ratio of SL and PF-68 added to the film could probably change the release rates of LY and AA. Furthermore, the utilization of SL in zein cast films served not only to improve the controlled release properties of films, but also support the anti-oxidant effects of films. The emulsified mechanisms indicated that the hydropathy property and the difference of molar weights in SL and PF-68 were the main reasons for the well-controlled release properties of the cast film. This study suggests the potential applications of the developed zein/surfactant composite films in food and pharmaceutics.

## Figures and Tables

**Figure 1 materials-15-02795-f001:**
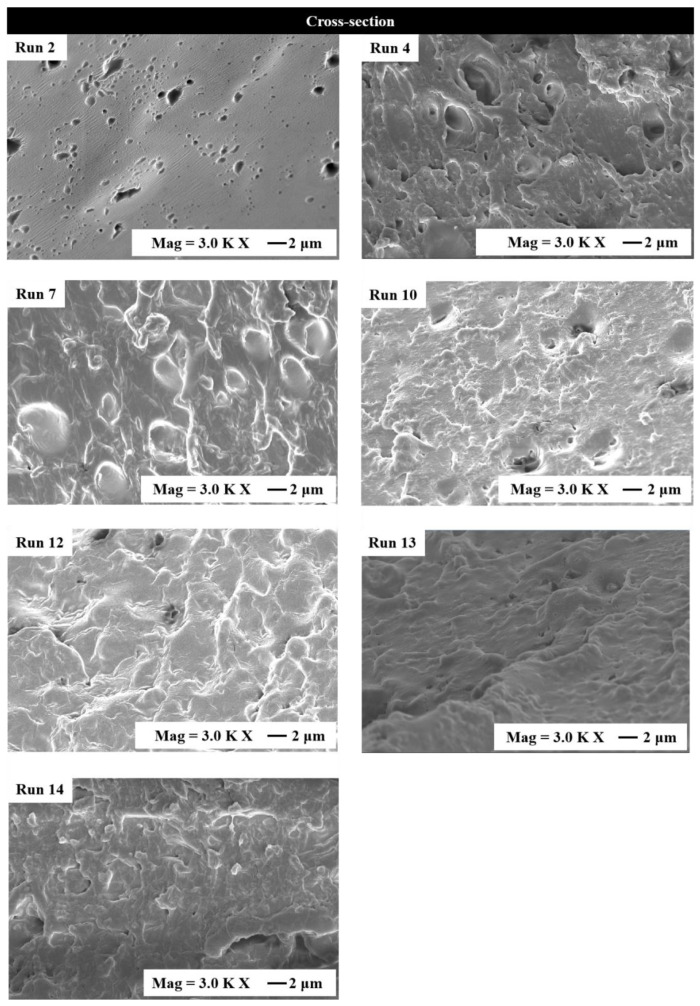
Cross-section SEM images of the typical zein/surfactant composite films.

**Figure 2 materials-15-02795-f002:**
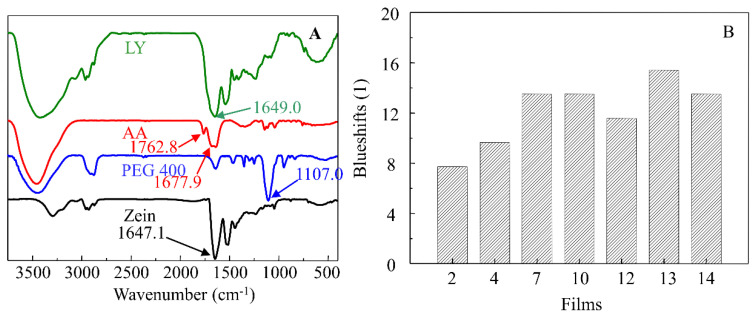
(**A**) main characteristic peaks of LY, AA, PEG 400 and zein; (**B**) blueshifts of the characteristic absorption of zein and LY at 1647.1 cm^−1^ in different zein cast films.

**Figure 3 materials-15-02795-f003:**
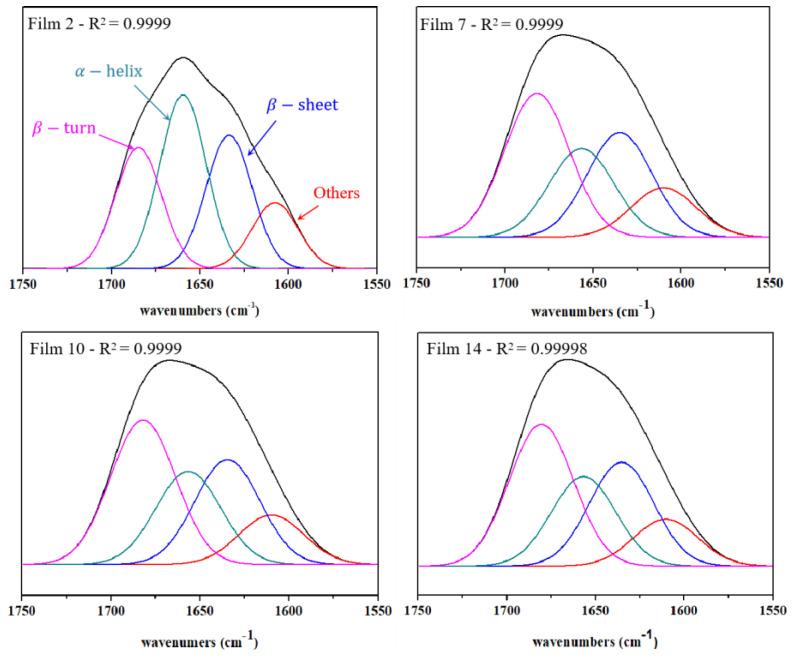
Deconvolution of the typical films using Peak Fit software according to the methods of Fourier self-deconvolution in the amide I region (1600–1700 cm^−1^) (Film 2, Film 7, Film 10, and Film 14).

**Figure 4 materials-15-02795-f004:**
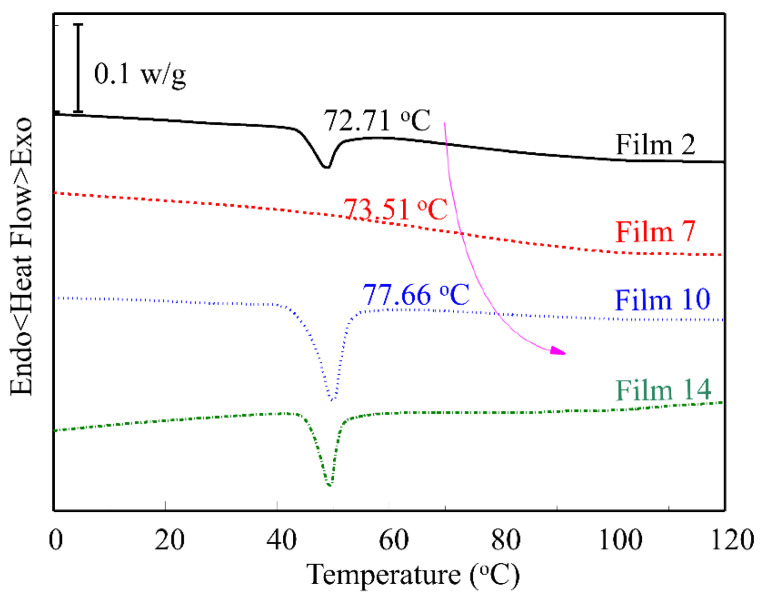
*T*_g_ change of zein-based films without surfactant (Film 2) and with surfactants of SL (Film 7), PF-68 (Film 10), and SL/PF-68 (Film 14). The arrow indicates the variation trend of *T_g_* in Films 2, 7, and 10.

**Figure 5 materials-15-02795-f005:**
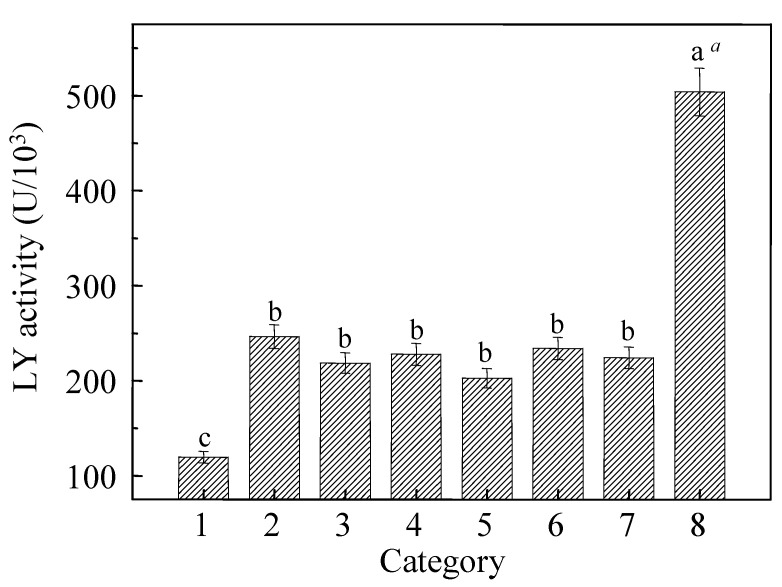
LY activities in different Et/H_2_O solutions. *^a^* Different letters (a, b and c) in each column show significant difference *p* < 0.05.

**Figure 6 materials-15-02795-f006:**
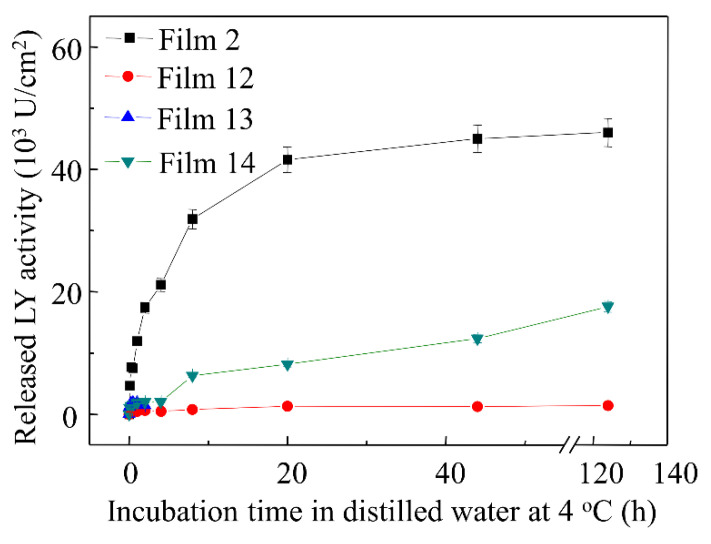
Released profiles of LY from zein cast films containing multiple surfactants.

**Figure 7 materials-15-02795-f007:**
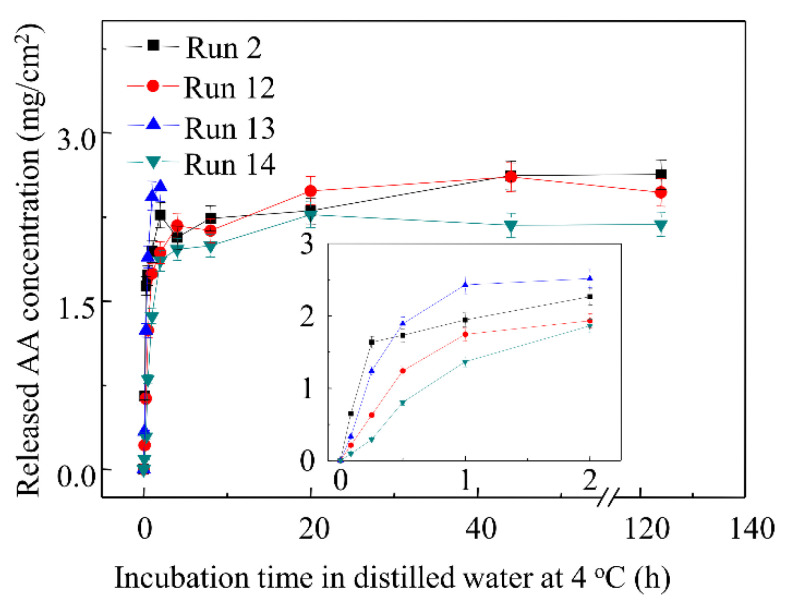
Released profiles of AA from zein cast films containing multiple surfactants.

**Figure 8 materials-15-02795-f008:**
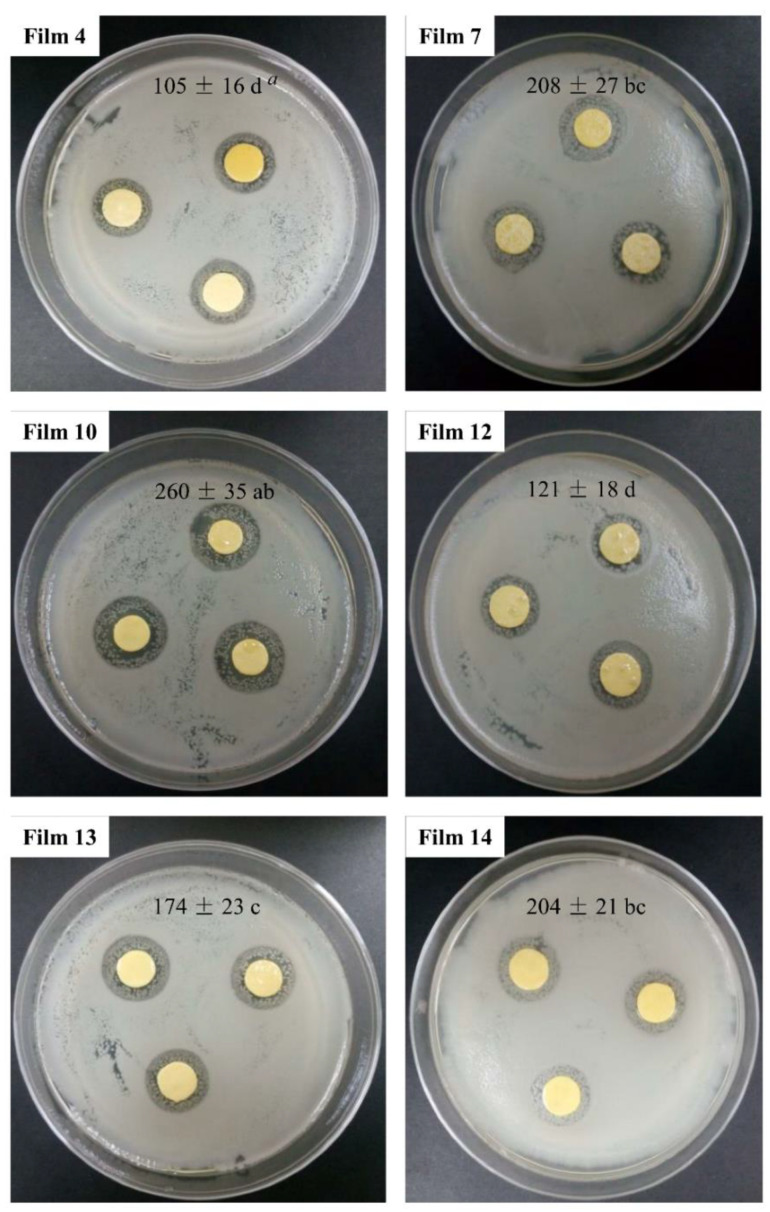
Anti-microbial potentials of zein/surfactants cast films against *L. innocua. ^a^* Different letters (a, b and c) in the figure show significant difference *p* < 0.05. Unit of the values is mm^2^.

**Figure 9 materials-15-02795-f009:**
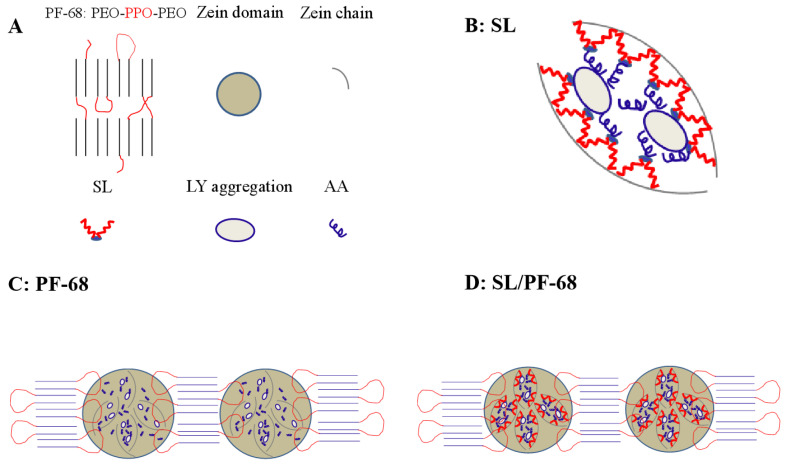
The emulsified mechanisms of SL and PF-68 crystallization on LY and AA in zein cast films. (**A**) Crystal structures of PF-68, zein domain in films, zein chain, SL, LY aggregation, and AA; (**B**) interaction of SL with LY aggregation, Zein chain and AA; (**C**) interaction of PF-68 with LY aggregation, Zein chain and AA; (**D**) interaction of SL-PF-68 with LY aggregation, Zein chain and AA. The blue and red color represents the hydrophilic and hydrophobic blocks/molecules respectively.

**Table 1 materials-15-02795-t001:** Positions and relative areas of the bands fitted to the Fourier- deconvoluted spectra of zein-based films.

Films	α−Helix		β−Sheet		β−Turn		Others	
	λ (cm^−1^)	A (%)	λ (cm^−1^)	A (%)	λ (cm^−1^)	A (%)	λ (cm^−1^)	A (%)
2	1659.6	35.1	1633.6	27.0	1684.9	24.5	1607.6	13.4
7	1656.6	22.9	1634.9	27.0	1681.9	37.2	1610.0	12.9
10	1656.6	23.7	1634.6	26.8	1682.1	36.9	1609.9	12.6
14	1656.8	23.4	1635.3	27.2	1680.6	37.1	1610.5	12.3

**Table 2 materials-15-02795-t002:** Mechanical properties of zein/surfactant composite films.

Film Composition *^a^*				TS (MPa)	EB (%)	Young’s Modulus (MPa)	Film Thickness (μm)
Films	SA (%)	SL (%)	PF-68 (%)				
1	-	-	-	1.65 ± 0.62 a *^b^*	30 ± 9 i	43.00 ± 21 a	167 ± 9
AA (3.1 mg/cm^2^) and LY (4.1 mg/cm^2^)							
2	-	-	-	1.02 ± 0.17 de	88 ± 8 efg	13.02 ± 0.15 fg	281 ± 20
3	5	-	-	0.96 ± 0.05 de	121 ± 19 bcd	6.84 ± 0.53 ij	354 ± 13
4	10	-	-	1.52 ± 0.05 ab	88 ± 21 efg	17.16 ± 2.76 e	295 ± 14
5	15	-	-	1.28 ± 0.08 bcd	141 ± 8 ab	11.04 ± 0.42 gh	312 ± 20
6	-	5	-	1.63 ± 0.10 a	59 ± 7 gh	26.51 ± 2.21 c	346 ± 30
7	-	10	-	1.39 ± 0.01 abc	71 ± 7.3 fg	20.33 ± 2.92 d	298 ± 9
8	-	15	-	1.57 ± 0.20 ab	32 ± 6 h	30.24 ± 2.44 b	340 ± 33
9	-	-	5	1.24 ± 0.06 bcd	132 ± 20 abc	7.46 ± 0.16 ij	317 ± 4
10	-	-	10	0.94 ± 0.07 de	129 ± 17 abc	9.20 ± 1.73 i	321 ± 16
11	-	-	15	0.78 ± 0.07 e	157 ± 25 a	5.02 ± 0.15 j	319 ± 5
12	5	5	-	1.28 ± 0.20 bcd	81 ± 17 efg	15.03 ± 3.46 ef	333 ± 12
13	5	-	5	1.01 ± 0.06 de	106 ± 28 cde	13.75 ± 1.75 efg	355 ± 23
14	-	5	5	1.07 ± 0.04 cde	99 ± 16 def	15.15 ± 0.47 ef	356 ± 15

Values reported were the means ± standard deviations. ***^a^*** All of the films contained 40% of PEG 400 as % of zein (*w/w*) and the concentration of SA, SL and PF-68 as % of zein (*w/w*). ***^b^*** Different letters in each column show significant difference *p* < 0.05.

## Data Availability

Not applicable.

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
