# Peer review of "Effects of Surfactants on Zein Cast Films for Simultaneous Delivery of Two Hydrophilic Active Components"

_materials, 2022, doi:10.3390/ma15082795_

Round 1

Reviewer 1 Report

Comments to the Author

MANUSCRIPT DETAILS

Ms. Ref. No.: materials-1660005

Title: Effects of Surfactants on Zein Cast Films for Simultaneous De-1 livery of Two Hydrophilic Active Components

Article Type: Research Article

Journal:  Materials

GENERAL COMMENTS

This manuscript aim of the current work is to develop zein/surfactant cast films for simultaneous 1 delivery of two hydrophilic active components, and investigate the effects of different 2 surfactants on the simultaneous release profiles of the model active components and 3 emulsified mechanisms of surfactants in the films.

The interest in this manuscript is significant enough to merit publication.

My recommendation on submitted manuscript to the materials journal is to be accepted after minor revisions.

The comments and questions provided below may help the authors to put the manuscript into better appropriate form for publication.

SPECIFIC COMMENTS

Abstract

Kindly specify clear and concise aim for your work, and the aim of Zein/surfactants cast films for simultaneous delivery of two hydrophilic ac-13 tive components were developed in Abstract sections.

Result and discussion

Page 6 line 1 Please add magnification (KV) and scale in SEM images

Page 7 line 27 Please add the letter of statistical analysis in table 1

Page 8 line 3 Please add a description of each film below the figure title  

Page 9 line 9 Please modify the letters of statistical analysis in table 2 (superscripts letters)

Page 13 line 2 Why was L. innocua strain used to determine antimicrobial activity of different films?

References

  • Revise references to be formatted in accordance with Journal of Toxins Author Guidelines.

Author Response

SPECIFIC COMMENTS

1. Abstract

Kindly specify clear and concise aim for your work, and the aim of Zein/surfactants cast films for simultaneous delivery of two hydrophilic ac-13 tive components were developed in Abstract sections.

A: Thanks for your kind advice. The aim for the work and the aim of zein/surfactants cast films for simultaneous delivery of two hydrophilic ac-13 tive components were added in Abstract and just as follows:”In order to prepare edible films with outstanding antimicrobials and antioxidants utilized in applications of food and pharmaceutics, in this study, effects of surfactants on zein cast films for simultaneous delivery of lysozyme (LY) and ascorbic acid (AA) were investigated, where sodium alginate (SA), soy lecithin (SL) and Pluronic f-68 (PF-68) were selected as surfactants.”

2. Result and discussion

2.1 Page 6 line 1 Please add magnification (KV) and scale in SEM images.

A: Thanks for your advice. The magnification (KV) and scale in SEM images on the Page 6 line 1 have been added and shown in Figure 1 in the revised manuscript.

2.2 Page 7 line 27 Please add the letter of statistical analysis in table 1.

A: Thanks for your advice. We are sorry to admit that because of our negligence, the letters of statistical analysis in table 1 should not exist owing to the tiny difference of the data. Thus, the sentence of “a Different letters in the column show significant difference P  0.05.” below the table 1 is deleted and the revision is shown in the revised manuscript.

2.3 Page 8 line 3 Please add a description of each film below the figure title.

A: Thanks for your advice. A description of each film below the figure title have been added and a description of each film in Figure 3 also have existed, which are displayed in the revised manuscript.

2.4 Page 9 line 9 Please modify the letters of statistical analysis in table 2 (superscripts letters).

A: Thanks for your advice. In fact, superscripts letters are only “a” and “b”, which describes the “Film composition” and “Different letters in each column show significant difference P  0.05” individually. While, the letters without superscript after the values in each column show the significant difference P  0.05, which should not be superscript.

2.5 Page 13 line 2 Why was L. innocua strain used to determine antimicrobial activity of different films?

A: Thanks for the question. L. innocua is a kind of primarily food-borne and one of the most deadly food-borne pathogens. According to data display, meat, eggs, poultry, seafood, dairy products and vegetables have all been confirmed as sources of listeria infection. Thus, in many studies, L. innocua strain used to determine antimicrobial activity of different films.

3.References

Revise references to be formatted in accordance with Journal of Toxins Author Guidelines.

A: Thanks for your advice. All the references have been formatted in accordance with Journal of Toxins Author Guidelines, which is displayed in the revised manuscript.

Reviewer 2 Report

The presented article is interesting, but the authors have shown negligence in presenting the material obtained, which at this stage reduces the value.

1. Page 2, line 16: What means "AR"?

2. Page 9: The authors should explain the effect of adding various surfactants on mechanical properties. For example, what can explain small changes in tensile stress and large changes in elongation break values?
Table 2: I would like to know the designations in more detail of "b-j".

3. Page 12, line 4: What means Figure 5 (A) ?

4. Page 12: Authors must be careful. In subsection 3.7, you refer to fig. 5, but the data are presented in fig. 6.

5. Page 12: No figure showing the effect of a single emulsifier on AA release profile.
Fig. 8 is Figure C4.

Author Response

The presented article is interesting, but the authors have shown negligence in presenting the material obtained, which at this stage reduces the value.

1. Page 2, line 16: What means "AR"?

A: Thanks for the question. We are sorry to make it unclear. “AR” is the abbreviation of “Analytic Reagent” and have been corrected and added in the revised manuscript.

2. Page 9: The authors should explain the effect of adding various surfactants on mechanical properties. For example, what can explain small changes in tensile stress and large changes in elongation break values?
Table 2: I would like to know the designations in more detail of "b-j".

A: Thanks for the advice. The reasons for the effect of adding various surfactants on mechanical properties have been added in Page 9 in the revised manuscript. The added segments are as follows: “The effect could be attributed to the hydrophilic property and long-chain structure of SA”, “ Because the hydrophilic head PC is higher than that of PEG 400, the addition of SL breaks the synergistic effects of PEG 400 and AA on strengthening the plastification of zein cast films” and “The dramatical increase of the EB values could be due to the hydrophilic segments of PF-68. ”

About the designations in more detail of "b-j" in Table 2, the detailed procedures are as follows: 1) download and install Statistical Product and Service Solutions software (SPSS); 2) set confidence interval, dependent variable and independent variable; 3)import and analyze data; 4) obtain the statistical analysis results.

3. Page 12, line 4: What means Figure 5 (A) ?

A: Thanks for the question. We are really sorry to admit the fault that “(A)” should not be added and have been deleted in the revised manuscript.

4. Page 12: Authors must be careful. In subsection 3.7, you refer to fig. 5, but the data are presented in fig. 6.

A: Thank you very much for the kind correction and it has been corrected in the revised manuscript.

5. Page 12: No figure showing the effect of a single emulsifier on AA release profile.Fig. 8 is Figure S4.

A: Thank you very much for the advice. The data showing the effect of a single emulsifier on AA release profile has been listed in Table S2. Figure S4 has been deleted and can been seen in the revised manuscript.

Reviewer 3 Report

The presented manuscript is devoted to the important topic of creating antimicrobial films for food storage. However, in the process of studying the manuscript, I noticed the following shortcomings.
1. The newest reference in the literature review and introduction is from 2017. Really for the last 5 years no one has dealt with films based on zein? Especially given the general trend towards green chemistry.
2. Section 2.3.2. Fourier transform infrared spectra. The authors should determine which spectrometer they used in the work Nicolet Nexus 670 or PE Paragon 1000. If different spectrometers were used for different studies within the work, it is necessary to indicate this in the captions to the figures.
3. Page 4. line 45 "For the film containing PF-68 (Film 10), a lot of nanoparticles..." reference 19 is given where crystallization processes are studied. However, only SEM images are presented in the work itself, without evidence of crystal formation. The authors should explain how they detected nano particles on their samples at 2 micron scale. In general, section 3.1. Microstructures at work seem redundant. Since it is a repetition of other works.
4. Figure 3. First, the x scale is incorrect. If you operate with wave numbers, they go from larger to smaller. The second does not show the convergence and standard deviation, compared with the original spectrum. third, water has an extremely strong signal at 1620 cm-1. Completely dehydrated protein samples were taken in vacuum? Otherwise, there could be no water signal.

Author Response

The presented manuscript is devoted to the important topic of creating antimicrobial films for food storage. However, in the process of studying the manuscript, I noticed the following shortcomings.
1. The newest reference in the literature review and introduction is from 2017. Really for the last 5 years no one has dealt with films based on zein? Especially given the general trend towards green chemistry.
A: Thanks for the advice. New references for last 3 years have been added and shown in the revised manuscript. The details are as follows:”[1] Mehdizadeh, A.; Shahidi, S. A.; Shariatifar, N.; Shiran, M.; Ghorbani-Hasansaraei, A. Physicochemical characteristics and antioxidant activity of the chitosan/zein films incorporated withpulicaria gnaphalodesl. extract-loaded nanoliposomes. J. Food Meas. Charact., 2022, 16, 1252–1262. [3] Coroli, A.; Romano, R.; Saccani, A.; Raddadi, N.; Mascia, L. An in-vitro evaluation of the characteristics of zein-based films for the release of lactobionic acid and the effects of oleic acid. Polymers, 2021, 13, 1826. [4] Federici, E.; Selling, G. W.; Campanella, O. H.; Jones, O. G. Thermal treatment of dry zein to improve rheological properties in gluten-free dough. Food Hydrocolloids, 2021, 115, 106629. [6] Zhao, Z.; Wang, W.; Xiao, J.; Chen, Y.; Cao, Y. Interfacial engineering of pickering emulsion co-stabilized by zein nanoparticles and tween 20: effects of the particle size on the interfacial concentration of gallic acid and the oxidative stability. Nanomaterials, 2020, 10, 1068. [7] Lza, B;  Kl, A.; Dya, B.; Jmrb, C.; Jda, B.; Wc, A. Chitosan/zein bilayer films with one-way water barrier characteristic: Physical, structural and thermal properties. Int. J. Biol. Macromol. 2022, 200, 378-387.”

2. Section 2.3.2. Fourier transform infrared spectra. The authors should determine which spectrometer they used in the work Nicolet Nexus 670 or PE Paragon 1000. If different spectrometers were used for different studies within the work, it is necessary to indicate this in the captions to the figures.
A:Thanks for the correction. We are so sorry to say that the Perkin Elmer spectrometer model Paragon 1000 was used in our previous work and now it has been deleted in the revised manuscript.

3. Page 4. line 45 "For the film containing PF-68 (Film 10), a lot of nanoparticles..." reference 19 is given where crystallization processes are studied. However, only SEM images are presented in the work itself, without evidence of crystal formation. The authors should explain how they detected nano particles on their samples at 2 micron scale. In general, section 3.1. Microstructures at work seem redundant. Since it is a repetition of other works.
A: Thanks for the advice and just as the reviewer says, section 3.1.at our work has a repetition of other works. However, the combined analysis of SEM and other characteristics will help to explain the release profiles of LY and AA and the emulsified mechanism. Thus, after our much cogitation, section 3.1 is still retained. The existence of nanoparticles with about 100-200 nm on the samples could be obtained from the Figure S1 (Film 10) in the “Supporting Information” 

4. Figure 3. First, the x scale is incorrect. If you operate with wave numbers, they go from larger to smaller. The second does not show the convergence and standard deviation, compared with the original spectrum. third, water has an extremely strong signal at 1620 cm-1. Completely dehydrated protein samples were taken in vacuum? Otherwise, there could be no water signal.

A: Thank you very much for the comments. Firstly, the x scale of the wave numbers have been corrected and go from larger to smaller. Secondly, convergence and standard deviation in Figure 3 have been added in the revised manuscript. Thirdly, before measurement next time, protein samples will be taken in vacuum and dehydrated completely.

Round 2

Reviewer 2 Report

The authors took into account all the comments of the reviewer. The article can be accepted for publication in present form.